# Deep Learning Projects from a Regional Council: An Experience Report

**Jónathan Heras** [1]

## Abstract

Due to the impact of Deep Learning both in industry and academia, there is a growing demand of graduates with skills in this field, and Universities are starting to offer courses that include Deep Learning subjects. Hands-on assignments that teach students how to tackle Deep Learning tasks are an instrumental part of those courses. However, most Deep Learning assignments have two main drawbacks. First, they use either toy datasets, that are useful to teach concepts but whose solutions do not generalise to real problems, or employ datasets that require specialised knowledge to fully understand the problem. Secondly, most Deep Learning assignments are focused on training a model, and do not take into account other stages of the Deep Learning pipeline, such as data cleaning or model deployment. In this work, we present an experience in an Artificial Intelligence course where we have tackled the aforementioned drawbacks by using datasets from the regional council where our University is located. Namely, the students of the course have developed several computer vision and natural language processing projects; for instance, a news classifier or an application to colourise historical images. We share the workflow followed to organise this experience, several lessons that we have learned, and challenges that can be faced by other instructors that try to conduct a similar initiative.

## 1. Introduction

Deep Learning (DL) techniques have become the state of the art approach to tackle problems in several domains such as computer vision (He et al., 2015), natural language processing (Devlin et al., 2019), recommender systems (Zhang et al., 2019), bioinformatics (Li et al., 2019) or games (Silver et al., 2018). Due to its success, there is a growing interest and demand of experts in this subfield of Machine Learning; and, several Universities are incorporating DL subjects in their courses.

In the intersected field of Data Science, studies on the design of Data Science courses have emphasised the importance of practical and application-based teaching (Song & Zhu, 2016; Ramamurthy, 2016), and this can be extrapolated to DL courses. In most DL courses today, this is achieved through assignments or projects where students have to train a DL model using a dataset acquired from public datasets portals (such as Kaggle, Amazon Datasets, or Google Datasets Search). However, the construction of a model is just one of the steps in the pipeline to tackle a project using DL — such a pipeline is summarised in Figure 1 — and the rest of the steps are not taken into account in most assignments. For instance, data acquisition, cleaning and labelling are not usually conducted by students since datasets from public portals are usually preprocessed and ready to be employed; or, models are not deployed since they are only evaluated by instructors, and they are not actually used outside the classroom. In addition, most publicly available datasets are either toy examples that have a limited interest for students, or require some knowledge of the field where the data was acquired. These problems can be approached by using open urban and regional data.

Many cities and councils around the world are investing a considerable amount of resources to publicly release their data (Silva et al., 2018). This opens the door to the application of Machine Learning to answer several questions of interest for citizens, administrators, businesses, and researchers. Building models using urban or regional data is close to a real project since such data must be usually cleaned; and to be useful for the society, models must be deployed. Moreover, students are familiar with the context of the data. In this paper, we present an initiative in an Artificial Intelligence course where students have developed, from end-to-end and using DL techniques, several computer vision and natural processing language projects suggested by a regional council. In addition, we introduce the tools and workflow employed to organise such an initiative. Finally, we conclude by presenting the challenges faced and the lessons learned during this experience.

*Equal contribution [1]Department of Mathematics and Computer Science, University of La Rioja, Spain. Correspondence to: Jónathan Heras <jonathan.heras@unirioja.es>.

*Proceedings of the 2nd Teaching in Machine Learning Workshop*, PMLR, 2021. Copyright 2021 by the author(s).

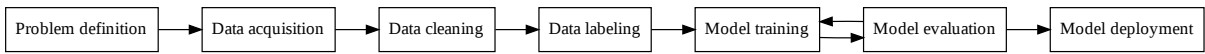

*Figure 1.* Pipeline of a Supervised DL project. After defining the problem to be solved, data must be acquired and cleaned. Subsequently, the data must be annotated to train a supervised prediction model. Once the model is constructed it must be evaluated using an independent test set that was not employed for training. After the model is evaluated, it must be deployed to facilitate its usage.

## 2. Experience description

This initiative was conducted in an undergraduate course on Artificial Intelligence. The goal of this experience was the development of DL-based solutions to projects suggested by the regional council where our University is located. The course involved 25 students, with either a Computer Science or Mathematics background, that worked in teams from 2 to 4 in a total of 9 projects, summarised in Table 1.

In the first step of this experience, instructors together with members of the regional council identified several tasks in a series of meetings. Those meetings were instrumental to find a set of sensible problems that could be handled in the context of the course; namely, we focused on projects with enough data that was, at least, partially annotated. Subsequently, a summary of the available projects was provided to the students, that formed teams and selected a project. After that, instructors created an assignment through GitHub classroom[1] that gave access to a private GitHub repository to each team. Each repository only contained a README file that explained how to access the data for each project, and some ideas and links explaining how to tackle the project.

The first task that the teams faced was the process of acquiring, cleaning and annotating the data. All the datasets were available through public APIs; however, the associated annotation was not usually in the same place. Hence, students had to match several sources of information. Moreover, most datasets were partially annotated; so, students had to clean them.

Once these preprocessing steps were conducted, students

---

[1]http://classroom.github.com

trained and evaluated their models using Python as programming language, and DL libraries like Keras (Chollet et al., 2015) or FastAI (Howard & Thomas, 2019). As development environment, they used Jupyter notebooks (Kluyver et al., 2016), an open-source and browser-based tool that allows the combination of text and code. Jupyter notebooks can be run locally; however, in order to train DL models, it is necessary the use of special purpose hardware like GPUs or TPUs, and most students do not have access to those resources. Hence, Jupyter notebooks were run in Google Colaboratory (Colab) (Nelson & Hoover, 2020), a pre-configured environment with the essential Machine Learning and DL libraries that provides access to free GPUs and TPUs through a Google account. It is worth mentioning that Colab can be linked with a GitHub repository; hence, students were able to easily save changes in their repositories. All the aforementioned tools were previously introduced to students during the course. The only feature that was unknown for them was the access of teams to a GitHub repository since the rest of the assignments in the course were individual.

After training their models, students evaluated them using an independent testing set that was not used for training the models. All models, except for the model in charge of colourising historical images, achieved over a 90% accuracy in their corresponding task — note that different metrics were employed for the different tasks. Hence, even if there is room for improvement, they can be considered as a success. Finally, students had to deploy their models in a way that they were easy to invoke. Most teams, 6 out of 9, decided to use forms in Colab; 1 team created a Desktop application, 1 created a web application; and the last team employed Binder[2].

In order to evaluate the assignments, we asked the students to document the whole process using Jupyter noteboks that should be stored in the teams' GitHub repository. Moreover, students had to present their work in a public exposition, and produce a 2-minutes video where they explained, in a non-technical manner, their work.

---

[2]https://mybinder.org/

| Project name | ♯ members | Tasks | Libraries | Deployment |
|---|---|---|---|---|
| People image retrieval | 4 | Face recognition | Keras | Desktop app |
| Colourising historical images | 3 | Image colourisation | FastAI | Colab |
| Colourising old aerial images | 3 | Image colourisation | FastAI | Colab |
| House detection in aerial images | 3 | Image segmentation | FastAI | Colab |
| News classifier | 3 | Text classification | FastAI | Colab |
| Classification of historical images | 2 | Image classification | FastAI | Colab |
| Dating historical images | 2 | Image classification | FastAI | Binder |
| Dating museum pieces from images | 2 | Image classification | FastAI | Web app |
| Classification of museum pieces | 3 | Image classification | FastAI | Colab |

*Table 1.* List of projects conducted during the experience

# 3. Lessons learned

In this section, we present the lessons that we have learned about how to organise similar experiences, and report the challenges that we faced and that should be taken into consideration when designing this kind of assignments. These recommendations are not only based on the instructors' opinion, but they are also founded on the students' satisfaction with the experience. To capture the students' satisfaction, we conducted an anonymous and non-compulsory survey developed with Google Forms. The survey consisted of 4 sections (valuation of GitHub, valuation of Jupyter notebooks and Colab, valuation of the projects, and comments). The valuation sections consisted of a list of questions following a 4-point Likert scale that goes from 1 (strongly disagree) to 4 (strongly agree), and the comments section allowed the students to introduce additional comments about the experience. The survey was answered by 15 out of the 25 students. We first describe the lessons learned from an organisational point of view.

**Regional data.** As suggested by previous work (Zeng et al., 2018), when considering assignments for undergraduate students, it is of great benefit to involve them in projects that are closely related to their life; hence, using regional data is a perfect match for this task. This fact can be seen in the video presentations where, for instance, the members of the team in charge of colourising historical aerial images showed the colourised image of the villages where their families live; or in the people image retrieval project, the members of the team searched themselves in the news of the regional council. In the anonymous survey, all the students claimed that they enjoyed working with real data from the regional council. In fact, one of the students wrote the following comment "*These assignments are great, they help you to apply what we have seen during the course in a real problem*".

**Meetings with members of the regional council.** In the long term, organising meetings with the members of the regional council has proven to be one of the best decisions taken for this experience. These meetings helped us to frame a set of problems that fit with the subjects explained in the course, solving the problem of finding enough and interesting data in urban datasets (Pineau & Bacon, 2015). In these meetings, instructors had to explain what could be done to the members of the council, and also establish some limits to their expectations — all the projects were proofs of concepts that had to be developed in a limited time, approximately 25 hours. In the following years, we plan to incorporate students into the loop of selecting the projects with the regional council.

**The README file.** Initially, each team had access to a GitHub repository with only a README file. In that file, instructors explained how to access the data, and pro-
vided some documentation explaining how to tackle the project. Among the documentation, we included links to open-source repositories containing Jupyter notebooks that tackle similar projects, and our first recommendation for the teams was to re-run those notebooks and use them as a basis for their projects. In this way, they had a starting point to know how to organise their datasets, and train their models. Moreover, the README file also contained pointers to additional research-oriented tasks; for instance, exploring several state-of-the-art architectures for image classification, or the application of ensemble methods. These research topics allowed some of the teams to dive deeper in the DL techniques.

**Videos.** We asked students to prepare short and non-technical videos explaining their work. The aim was twofold. First, students had to make an effort to summarise their work in appealing manner. And, secondly, these videos served to present the work to members of the regional council the kind of tasks that can be solved using DL techniques. This is important to continue this experience in the future with new projects.

Now, we focus on the lessons learned from a technical point of view. These lessons can be applied not only to projects based on open-data but also to any DL project.

**Transfer learning.** Training a DL model from scratch is a time-consuming task that requires lots of data and computational resources; and, in most cases, it is not feasible to access to such amount of resources. This problem can be faced by applying a widely used technique known as transfer learning (Razavian et al., 2014); a method that reuses a model trained in a source task in a new target task. This considerably reduces the amount of data and time that is needed to construct an accurate DL model. This technique can be employed in almost any DL project; hence, it is important to introduce it throughout the course; so, students can employ it in their assignments.

**GitHub and GitHub classroom.** In this experience, all the code and models generated from the assignments were stored in GitHub repositories. Namely, all the repositories belong to an "organization" managed by the instructors of the course. In this way, students only have access to their team's repositories, and instructors have access to all the students' repositories. This approach not only simplifies the collaboration of teams' members, but also improves the communication with instructors, since they can directly access the code of students when they need feedback. In addition, instructors can follow the progress of students since it is possible to monitor the commits and who made them. From the students point of view, the usage of GitHub had positive valuations from students. Most students thought that GitHub helped them to manage better the code of their assignments, and considered that it was easy to use and

facilitated working on teams.

**Colab.** This environment has proven to be enough to train DL models for the assignments. In addition, all of the surveyed students were satisfied with the use of Colab, and considered that it was well integrated with GitHub. An issue that arose when using Colab is that it has a 12 hours limitation (that is, after 12 hours, the environment halts); hence, students have to save intermediate checkpoints to avoid loosing their work, since in some cases 12 hours were not enough to complete the training process. It would be possible to train for longer using cloud environments like Amazon AWS or Google Cloud, but their configuration and management are more difficult, require adding billing information, and are out of the scope of the course.

**Updates of libraries.** DL is a field in constant evolution, and there is a continuous update of libraries implementing DL methods. This can lead to code that works for a version of a DL library, but fails when the library is updated. This can be solved locally by using virtual environments where the versions of the libraries are fixed by the user. However, Colab updates regularly the version of the libraries included in the underlying virtual machine; and, this is a problem for two reasons. First, students found out that their code stopped working from one day to another; and, second, the code from the online tutorials that were used as a basis raised errors when executed. In order to tackle this challenge, we provided the students with a requirements file that can be used to install concrete versions of a set of libraries in Colab.

**FastAI as DL library.** Nowadays, there are two main DL frameworks employed both in industry and academia: Tensorflow and Pytorch (He, 2019). However, for projects like the ones presented in this experience, it is better to use a library that provides a straightforward access to different kinds of models, and also implements best practices. During the course, we presented two of those libraries: Keras (Chollet et al., 2015) and FastAI (Howard & Thomas, 2019). For the assignments, teams could choose the library to employ, and all but one used FastAI. This was mainly due to the user-friendly API of this library that allows users to train state-of-the-art models with just a few lines of code.

**Data management.** The last lesson is related to how teams managed their data. In the Colab environment, data is only kept during the session; hence, teams had to upload their data every time. To deal with this issue, we asked students to create a Jupyter notebook devoted only to download and clean the data, and once the data was processed, they had to upload the data to a file hosting system. In this way, both the members of the team and the instructors could easily access to the clean version of their data.

We focus now on the challenges that should be taken into account when this kind of experience is conducted.

**Instructors' load.** One of the main challenges of this experience was the load for instructors both in terms of organisation of the assignments and supervision of the teams. Therefore, instructors must have the time to organise the assignments (that is, contact and meet with the members of the regional council, and prepare the materials to guide the students) and also provide individualised advice to each project team. In addition, in this kind of assignments, it is difficult to foresee the difficulties that students will face; and, this means an additional load in terms of supervision.

**Reproducing errors.** A challenge related to the supervision is how to reproduce errors found by the teams. One of the greatest advantages of GitHub repositories and Colab is the chance of accessing teams code; and, therefore, it is easy for instructors to inspect the error messages. However, just inspecting the error messages might not be informative enough, and it might be necessary to re-run the code. This might be challenging since, in some cases, it requires running a lot code before reaching the point where students encountered the problem; and, in addition, Jupyter notebooks' hidden state make difficult to reproduce the exact conditions where students had their problems (Grus, 2018). In order to minimise this challenge, our recommendation for students was to create a devoted notebook for each training experiment; in this way, it was easier to provide them with feedback to tackle their problems.

**Methods beyond the curriculum.** 4 out of the 9 projects required students to apply DL methods that were not explained during the course; and, the other 5 projects included additional tasks for diving deeper in the topics of image and text classification. To deal with this issue, we pointed the students to materials with examples and explanations of the techniques to employ. However, a problem that we found with this approach is that students focused on just training the models and did not understand the underlying methods. A solution to tackle this problem is based on the work of the team in charge of segmenting houses from aerial images. This team created a notebook explaining the U-net segmentation architecture (Ronneberger et al., 2015), and provided a toy example explaining how to use it. In the future, we will explore this approach.

## 4. Conclusions

We have presented a initiative to apply DL methods to create solutions to regional projects. In this experience, students have worked with open data that is familiar for them, and they were involved in all the stages of the pipeline to develop a DL project. The goal of this paper was to present the lessons learned and challenges faced to organise similar initiatives. Among those lessons, it is worth highlighting the benefits of involving members of the regional council to select a set of projects that motivate the students.

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
