# OpenReview forum: "Deep Learning Projects from a Regional Council: An Experience Report"
_ecmlpkdd.org/ECMLPKDD/2021/Workshop/TeachML — TeachML 2021_

### Official Review · Reviewer_ZX2b · 2021-07-09
**Using local datasets in projects**

**Rating:** 6
**Confidence:** 4

**Review:**

This paper described projects given as part of an undergraduate course in AI. The authors identified two problems with existing project structures:
1) Using toy datasets that do not generalize to real world problems (or, datasets that require a lot of background knowledge to understand)
2) Projects do not include data cleaning and model deployment.

Their solutions are 1) to use datasets from a regional council, that require cleaning and alignment, and 2) require that the model is deployed in an easy-to-invoke manner.

Things I liked:
* The use of local data makes the project relevant to the students
* Explicit focus on data pre-processing
* Clear framework for model development
* Consideration of limitations of jupyter + colab (time limits, challenges debugging)

Things I was disappointed by:
* One of the two goals (OK, half of one of the two goals) was that the project focused on model deployment. However, this was barely discussed in the paper. What standards were required for model deployment? How much additional work did the students have to spend here?
* The local datasets were not discussed in huge detail---the level of detail was fine for understanding the course as a whole, but given the main motivation of the paper was the datasets, I expected to see more discussion.
* I would gently push back on "toy datasets don't require cleaning and don't generalize". There are plenty of messy datasets on kaggle, and it's unclear exactly what the authors are referring to by toy datasets not generalizing. This is only a minor critique, because I do like using local datasets and there is a clear advantage to having datasets that are not already on kaggle. I would just like to see more concrete discussion of why they are preferred over alternatives.
* Beyond sourcing the dataset, the model development framework seems fairly standard (using notebooks, using github, using colab). I liked that the process was well-detailed, but there wasn't much that I haven't seen in existing courses.
* The authors mention that they gathered feedback, but don't include a summary of that feedback. It would be great to see how the feedback compared with a previous version of the course that didn't include the local data and the deployment, if such data exists, or more generally how it compares with other ML course evaluations.

Overall, I found this a clear description of the course setup and it is clear that the authors have been thoughtful about their technological choices, but didn't find anything particularly innovative.

---

### Official Review · Reviewer_DkyD · 2021-07-12
**Interesting experiment, but with a few remaining questions to be addressed**

**Rating:** 6
**Confidence:** 5

**Review:**

Paper is clear and well written. However, it is 5 pages (only references on the last page, but the page is almost full of them, so reducing the size will be difficult...)

Paper exposes challenges & lessons learned while making an experiment to make students work on actual datasets/problems, so that then not only learn to train Machine Learning models, but also to clean datasets and deploy the models. Solving such actual problems is also more motivating/rewarding for students. Problems were built in collaboration with the regional council.

The problem addressed in this paper is indeed one of the main challenges of teaching Machine Learning (ML)/Data Science (DS): it is well known that acquiring/cleaning/preprocessing data is 90% of the job of Data Scientists. But during teaching, time is limited. So teachers generally either:
1- use "toy" datasets, so that the students do not have to clean the data & can focus on understanding the inner working of the models and ML process. But in the end students are totally lost when they get hired by a company and have to work on actual data. Or,
2- use "actual" datasets, thus the students learn the important craft of data cleaning/preprocessing but may waste many hours in acquiring/cleaning the data and loose focus on ML models.

The approach taken here is thus very interesting. However, in the end it falls in the pitfalls of approach 2): it seems that, in the end, the students mostly did transfer learning, by adapting existing Deep Learning models. Also they almost all use FastAI because "[it] allows users to train state-of-the-art models with just a few lines of code". So, in the end, students did not learn the crafts of Designing/Tuning/Evaluating ML Models. Is that really what we want to teach to ML/DS students? In the same manner, I'm not convinced writing a "Google Form" could be considered a "model deployment".

Also, the "Lessons learned" section lists a lot of problems faced due to the use of proprietary cloud solutions (Google Colab / GitHub / Binder / Google Forms / FastAI): reset every 12 hours, data loss, etc? and the not-so-simple workarounds that had to be found by the teachers & students. Do these problems actually balance the fact that directly using the students' machines (e.g. with tools installed locally) might have been slower (for those students with no big graphics card)? Also, learning how to install/configure the tools you use is, IMHO, one of the craft that any DS should have. If one does not know how one's tools work under the hood, then one might make very big mistakes when using them, particularly with tools that manipulate data or configure ML models.
If the teachers really wanted the advantages of a cloud platform but without the limitations of the proprietary solutions, why didn't they installed a server with FLOSS like Jupyter/Git/GitLab in their university? Such an alternative solution that would have had all advantages of both solutions...

---

### Decision · Program_Chairs · 2021-07-21

**Decision:**

Accept

**Comment:**

Congratulations! The reviewers agree that this paper should be accepted.

Camera-ready version is due August 18, 2021. As you prepare the camera ready version, please take the reviewers comments into consideration.

We look forward to your participation at the workshop on September 13, 2021. We invite you also to join us for the satellite event on September 08, 2021. Schedules for both the workshop and the satellite event will be forthcoming.